# Insights into Macrophage/Monocyte-Endothelial Cell Crosstalk in the Liver: A Role for Trem-2

**DOI:** 10.3390/jcm10061248

**Published:** 2021-03-17

**Authors:** Inês Coelho, Nádia Duarte, Maria Paula Macedo, Carlos Penha-Gonçalves

**Affiliations:** 1CEDOC, NOVA Medical School, Faculdade de Ciências Médicas, Universidade Nova de Lisboa, 1150-082 Lisboa, Portugal; ines.c.coelho@nms.unl.pt (I.C.); paula.macedo@nms.unl.pt (M.P.M.); 2Instituto Gulbenkian de Ciência, 2780-156 Oeiras, Portugal; nduarte@igc.gulbenkian.pt; 3APDP Diabetes Portugal, Education and Research Center (APDP-ERC), 1250-189 Lisbon, Portugal; 4Department of Medical Sciences, Institute of Biomedicine—iBiMED, University of Aveiro, 3810-193 Aveiro, Portugal

**Keywords:** macrophages, endothelial cells, liver disease, cell interactions, Trem-2

## Abstract

Liver disease accounts for millions of deaths worldwide annually being a major cause of global morbidity. Hepatotoxic insults elicit a multilayered response involving tissue damage, inflammation, scar formation, and tissue regeneration. Liver cell populations act coordinately to maintain tissue homeostasis and providing a barrier to external aggressors. However, upon hepatic damage, this tight regulation is disrupted, leading to liver pathology which spans from simple steatosis to cirrhosis. Inflammation is a hallmark of liver pathology, where macrophages and endothelial cells are pivotal players in promoting and sustaining disease progression. Understanding the drivers and mediators of these interactions will provide valuable information on what may contribute to liver resilience against disease. Here, we summarize the current knowledge on the role of macrophages and liver sinusoidal endothelial cells (LSEC) in homeostasis and liver pathology. Moreover, we discuss the expanding body of evidence on cell-to-cell communication between these two cell compartments and present triggering receptor expressed on myeloid cells-2 (Trem-2) as a plausible mediator of this cellular interlink. This review consolidates relevant knowledge that might be useful to guide the pursue of successful therapeutic targets and pharmacological strategies for controlling liver pathogenesis.

## 1. Introduction

Being the largest solid organ in the human body, the liver is classified as an accessory digestive organ, but its multiple roles, namely, regulation of whole-body energy metabolism, makes it vital. Given its positioning in the blood circulatory system, the liver also possesses the unique characteristic of receiving all materials and compounds adsorbed from the intestine. This turns it into a preferred site for detoxification and for immune response against invading microbes. Furthermore, the liver also senses antigens from systemic circulation via its arterial blood supply [1].

The liver is structurally and functionally heterogeneous. Hepatocytes, the parenchymal cells, are the most numerous cells of the liver, comprising 60% of the organ’s total cells and 80% of its volume. Non-parenchymal cells (NPCs) represent around 40% of liver cells [2] and include sinusoidal endothelial cells (LSECs) (20% of liver cells), Kupffer cells (KCs) (approximately 15% of liver cells), and hepatic stellate cells (HSC) (5–8% of liver cells). Other immune cell populations, mainly natural killer T cells (NKTs) comprise a minor fraction of NPCs [3]. Hepatocytes perform the majority of liver functions, however, liver macrophages also play critical roles to maintain organ and body homeostasis [4,5].

## 2. Macrophages in the Liver

Macrophages were first described by Eli Metchnikoff in the late 19th century, who observed phagocytic activity and the participation of these particular cells in the maintenance of tissue homeostasis and integrity [6,7]. With a high phagocytic capacity, macrophages are fundamental for an organism’s tissue development, repair, to immune response against pathogens and environmental aggressors. Macrophages have been classified as belonging to the mononuclear phagocytic system (MPS) [8], which consists of highly phagocytic cells (professional phagocytes) and their bone-marrow progenitors. Nevertheless, monocytes and macrophages have distinct functions in tissue homeostasis and immunity. While monocytes are key players in inflammation, tissue resident macrophages act mostly during tissue development and homeostasis, tissue repair, and resolution of inflammation [6].

### 2.1. Liver Resident Macrophages—Kupffer Cells

Kupffer cells (KCs) are the resident macrophages of the liver comprising the largest tissue-specific population of macrophages in the body (80–90% of tissue macrophages [9]. The major immune function of KCs in healthy liver is linked to phagocytosis of microbes and other particulate materials and the presentation of antigens entering the liver from portal vein or arterial circulation. Given their anatomical position, KCs constitute one of the front lines of defense against pathogens that cross the intestinal barrier. Accordingly, KCs are equipped with scavenger, complement, and pattern recognition receptors (PRRs) (e.g., toll-like receptors: TLR4 and TLR9), enabling innate recognition of a large spectrum of antigens including pathogen-associated molecular patterns (PAMPs) and damage-associated molecular patterns (DAMPs) [10]. Responses upon antigen uptake by KCs show mostly a tolerogenic phenotype, preventing undesirable immune responses to the common gut-derived antigens that they are constantly in contact with.

Although general markers such as F4/80 are used to identify tissue macrophages [11], specific ones can be employed to identify KCs. Among these are Clec4f [5], [12,13,14] and Tim4 [12], which are expressed in other tissue macrophages [15] but not in monocyte-derived cells. Compared with resident KCs, liver macrophages can be increased through macrophage differentiation from newly recruited monocytes that present distinct antimicrobial and inflammatory capacity [16]. It is still debatable whether these recruited cells form a fully separate population independent of liver self-renewed macrophages from fetal origin.

In addition to their immune function, KCs have pivotal functions in iron and lipid metabolism in the liver. Bulk transcriptomic analyses showed that proteins involved in iron and lipid metabolism are highly expressed in KCs [12]. KCs are responsible for recycling and clearing senescent and dying erythrocytes, thereby providing further control of systemic iron levels [17,18]. KCs are unable to cope in cases of iron overload, and consequently die by a specific process entailing iron-mediated oxidative stress and lipid peroxidation, called ferroptosis [19]. Notably, a link between the immune function and iron metabolism has been shown, with increased KCs iron pools being simultaneously advantageous to certain microbes and promotor of changes in the KC transcriptional profile that limit macrophage capacity in killing bacteria [17].

### 2.2. Macrophages in Liver Disease

#### 2.2.1. Macrophages and Non-Alcoholic Fatty Liver Disease (NAFLD)

Non-alcoholic fatty liver disease (NAFLD) is the most common liver disorder in Western industrialized countries and has a reported prevalence of 6–35% globally. In Europe, the median prevalence is 25–26% with large variations in different populations [20]. NAFLD is an asymptomatic condition that reflects the epidemiological impact of wide consumption of high-energy diets. Nevertheless, NAFLD is now recognized as a major cause of end-stage liver failure [14,21].

The hallmark of NAFLD is an excess of fatty acids and lipids in the liver that results in lipotoxicity and hepatocyte injury initiating tissue inflammation [4]. A substantial number of NAFL cases develop a more inflammatory disease subtype, non-alcoholic steatohepatitis (NASH), that in turn may progress to severe liver fibrosis or cirrhosis—the major determinant of liver-related morbidity and mortality [21]. NAFLD pathogenesis entails multiple intrahepatic and extrahepatic events that concur to disease evolution. Intensive research in recent years has revealed a critical role of innate immunity in disease progression [22].

In NAFLD progression, Kupffer cells are the first immune cells to be activated followed by monocyte-derived recruited macrophages. Kupffer cells are equipped to ingest lipids and digest them in the lysosome, thereby generating cholesterol and fatty acids [11,23]. The involvement of KCs in NAFLD is illustrated in a model of steatosis induced by high-fat diet (HFD) where KCs accumulate toxic lipids, resulting in altered expression of genes involved in lipid metabolism and in pro-inflammatory mediators production [24]. Interestingly, despite this activation state, KC numbers were not different between steatotic and normal livers.

In the context of NAFLD, macrophages with a pro-inflammatory phenotype seem to contribute to disease severity. Macrophage infiltration in the liver has been demonstrated in mouse dietary models such as HFD and methionine- and choline-deficient (MCD) diet [25,26]. A mouse model of NASH induced by high-fat and high-cholesterol (HFHC) diet, which replicates pathophysiological features of human NASH, displays F4/80+ macrophages infiltration and activation of KCs with increased expression of pro-inflammatory cytokines [27]. Remarkably, in a study where mice were fed HFD and KCs were depleted using clodronate-containing liposomes, it was demonstrated that ablation of the resident macrophage compartment leads to decreased steatosis [28]. In accordance, ablation of KCs ameliorates steatohepatitis and is linked to decreased expression of MCP-1 and to diminished Ly6C+ monocyte recruitment to the liver [29]. Moreover, it was reported that selective depletion of KCs using gadolinium chloride could ameliorate hepatocyte insulin resistance in rats after 2 weeks of high-fat or high-sucrose diet [30]. Together, these findings support the notion that KCs have a pro-inflammatory role in the initial stages of NAFLD development.

Several studies have provided evidence that monocyte-derived macrophages contribute to the pathogenesis of NAFLD by worsening tissue inflammation and disease prognosis. In experimental genetic (ob/ob mice) or diet-induced (HFD feeding) mouse models of obesity, hepatic expression of CCR2 was associated with body weight gain and with higher numbers of recruited monocytes showing an increased pro-inflammatory profile when compared with resident Kupffer cells [31]. In a study of diet-induced obesity where Kupffer cells and blood-derived monocytes were differentially labelled in vivo, the number of recruited monocytes with a pro-inflammatory phenotype was increased in obese when compared to lean mice [32]. Monocyte-derived macrophages’ involvement in NAFLD is further indicated by the finding of increased infiltration of Ly6C+ CD11b+ monocytes differentiated into pro-inflammatory macrophages in HFD murine livers [33]. Similar findings were recapitulated in a human NAFLD study, where CCR2 was found to be highly expressed in monocyte-derived macrophages and where NAFLD severity was positively associated with CCR2+ macrophages in obese patients [34,35].

#### 2.2.2. Macrophages and Chronic Liver Disease

Epidemiological data estimate that chronic liver disease (CLD) afflicts approximately 180 million people worldwide. Major etiologies of CLD are: (1) chronic infections, e.g., hepatitis; (2) chronic exposure to toxins or drugs, e.g., alcohol; (3) impaired metabolic conditions, associated with type 2 diabetes and obesity, and (4) persistent autoimmune injury [36]. Over the last years, liver fibrosis, a common pathological feature of CLD, has become an increasing health concern [37].

Irrespective of the initial causes of injury, the fibrogenesis process is similar to and shares common pathways of chronic hepatic inflammation [38]. Progression to cirrhosis, the end-stage of liver fibrosis, results from sustained wound healing of chronic injury and consequent scar tissue formation with loss of tissue architecture and ultimately organ failure [38]. Still, the liver has a high regeneration capacity [39]. As such, liver fibrosis can be reversed when the injury is removed and repair responses dominate [38].

Macrophages are central players in the development of liver fibrosis [40]. Recent reports from phase 2 clinical trials in patients with non-alcoholic steatohepatitis (NASH) showed reduced liver fibrosis when targeting macrophages using the inhibitor of the serine/threonine kinase ASK1, selonsertib [41], or the dual CCR2/CCR5 inhibitor cenicriviroc [42].

Responses to liver damage are initiated by release of cell contents and inflammatory mediators that activate Kupffer cells and hepatic stellate cells (HSCs). Activated resident macrophages (KCs) secrete pro-inflammatory mediators such as TNF-α, IL1-β, and CCl2/CCl5, which all mediate recruitment and homing of bone marrow monocytes to the site of injury. Similar to the clinical setting of acute liver injury, throughout the fibrotic response KCs are virtually undetectable and give rise to recruited monocyte-derived cells that become the predominating liver macrophage population. Recruited macrophages show highly pro-fibrotic and pro-inflammatory profiles and secrete inflammatory mediators (e.g., TNF-α and IL1-β) that may further aggravate hepatocellular damage. On the other hand, TGF-β and PDGF produced by activated recruited macrophages are key activators of quiescent HSCs and inducers of their transdifferentiation into myofibroblasts. Activated myofibroblasts synthesize α-smooth muscle actin (α-SMA) and type I collagen, thereby leading to excessive accumulation of extracellular matrix (ECM) [39] and originating the fibrosis phenotype.

## 3. Liver Endothelial Cells

Cells from the endothelial lineage are involved in promoting organogenesis during normal embryonic development. In the context of tissue and organ regeneration, it is widely accepted that endothelial precursors generate a trophic microenvironment for a variety of tissue-specific cell types thereby playing a direct role in launching vascular niches [43]. The existence of tissue-resident vascular endothelial stem cells (VESCs) has been a matter of intense debate in the field of angiogenesis. It was recently proposed that VESCs residing in organ blood vessels, namely, in the liver, are capable of clonal expansion and subsequent blood vessel formation during normal physiologic turnover and after liver injury [44]. Furthermore, a specific subset of liver sinusoidal endothelial cells (LSECs) was demonstrated to initiate and sustain liver regeneration after hepatectomy [43], while bone-marrow (BM)-derived sinusoidal endothelial cells (BM-SPCs) were identified as drivers of liver regeneration [45]. It was demonstrated that these cells have the ability to restore loss of LSECs after injury or partial hepatectomy [46,47].

### 3.1. Liver Sinusoidal Endothelial Cells (LSECs)

Liver sinusoidal endothelial cells (LSECs) are highly specialized endothelial cells that form the wall of the liver sinusoids and represent approximately 15–20% of liver cells and 3% of the total liver volume. They have a discontinuous architecture, with areas called “fenestrae” acting as dynamic filters that make liver sinusoids highly permeable [48]. Under physiological conditions, LSECs are quiescent and have a low proliferation rate and long-life span, similar to endothelial cells from large vessels [48]. LSECs reside at an interface between the sinusoidal lumen and the space of Disse [49]. On the sinusoidal side, they are exposed to the highly oxygenated arterial blood mixed with portal blood derived from the gut and pancreas containing nutrients, bile acids, and hormones insulin and glucagon. On the abluminal side, they interact with hepatic stellate cells and hepatocytes [48]. Further, LSECs serve as the liver’s main source of nitric oxide (NO) via endothelial nitric oxide synthase (eNOS) activation and also maintain hepatic stellate cell quiescence [50]. In addition, they can be activated and show potent immunological functions [51].

### 3.2. Liver Sinusoidal Endothelial Cells (LSECs) and Liver Disease

Loss of LSEC fenestrae, known as capillarization, along with LSEC dysfunction occur early in NAFLD [52,53], thereby compromising their ability to act as vasodilator agents and to respond to shear stress [54]. In fact, it has been illustrated that capillarization arises from simple steatosis up to the early NASH stage and worsens in a time-dependent manner [52], setting the stage for steatosis development and NAFLD progression.

The release of inflammatory mediators by LSECs further sustain liver injury and inflammation. During NASH, LSECs progressively overexpress adhesion molecules such as ICAM-1, VCAM-1, and VAP-1 [55,56,57], as well as pro-inflammatory mediators such as TNF-α, IL-6, IL-1, and CCL2 [55,57,58]. Under lipotoxic conditions, Fas receptor, which is normally absent, is expressed on LSECs and binds to activated immune cells expressing Fas ligand [51,59].

In chronic conditions, in addition to sinusoid capillarization, LSECs mediate angiogenesis and secrete angiocrine factors. Chronic injury causes persistent FGFR1 activation in LSECs, perturbing the CXCR7-Id1 pathway and favoring a CXCR4-driven pro-fibrotic angiocrine response, ultimately promoting liver fibrosis [60]. In organ fibrosis, it was determined that, in livers derived from cirrhotic patients and mice treated with CCl4, a subpopulation of liver endothelial cells undergoes endothelial to mesenchymal transition (EndMT) [61]. EndMT consists of a cellular transdifferentiation defined by loss of cellular adhesion, acquisition of invasive and migratory properties, and cytoskeletal reorganization [62,63]. Liver endothelial cells from healthy mice were able to transition into a mesenchymal phenotype in culture in response to TGF-β1 treatment. Fibrotic livers treated chronically with BMP7, a potent EndMT inhibitor [64], showed lower EndMT acquisition and a positive impact on disease severity via reduced fibrosis and reduced sinusoidal vascular disorganization [61]. EndMT can be induced by multiple autocrine and paracrine signaling molecules produced due to tissue injury or by immune cells recruited to the site of injury in response to inflammation [65]. A number of inflammatory mediators, including pro-inflammatory cytokines, growth factors, oxidative stress and toxins, are able to induce the conversion of endothelial cells into mesenchymal fibroblast-like cells that sustain disease progression [66].

Moreover, during NASH and liver fibrosis, LSECs fail to maintain KCs and hepatic stellate cell quiescence, resulting in KCs activation, which, in turn, maintains liver injury and potentiates liver fibrosis from recruitment and adhesion to leukocytes [54,67].

## 4. Macrophages and Endothelial Cells Crosstalk

### 4.1. Macrophages and Endothelial Cells in Homeostasis

While past studies have often focused on analyzing particular cell populations, it is now well recognized that liver organization comprises tightly regulated mechanisms coordinated by the interactive crosstalk of different cell types. Recently, these interactions have received greater attention as novel specific programs were discovered to participate in this precise regulation.

Under homeostatic conditions, endothelial cell and leukocyte interactions are closely controlled by local environment cues. Endothelial cells secrete nitric oxide (NO) and prostacyclin (PGI_2_) in order to promote vascular integrity [68] and liver resident macrophages (KC) are mostly tolerogenic. In healthy conditions, cell-to-cell interactions in the liver are mostly associated with maintenance of each specific cell niche (Figure 1). Among others, the mechanisms by which tissue resident macrophages (KC) acquire resident phenotypes have been a matter of growing investigation. It has been shown that upon acute KC depletion, combinatorial crosstalk from sinusoidal endothelial cells is needed to induce and maintain KC identity. In a particular study, KCs were ablated using transgenic mice that expressed diphtheria toxin receptor (DTR) on a KC specific gene, Clec4f. This resulted in rapid colonization of the empty niche by circulating monocytes and their differentiation to KC [69]. The authors found that a combinatorial and sequential model of differentiation of myeloid progenitors to KCs involves the LXR, Notch, and TGF-β family signaling pathways—a critical initial step for the interaction of recruited monocytes with delta like canonical notch ligand 4 (DLL4) on LSECs. Moreover, Notch signaling appeared to simultaneously initiate a program of KC gene expression as well as suppress the expression of monocyte-specific genes. This work provides key novel insights into how a common macrophage progenitor acquires tissue-specific phenotypes (Figure 1). Interestingly, another study also showed that KC loss, again using a transgenic mouse expressing DTR on Clec4f, leads to induction of TNF and IL1 receptor activation of stellate cells and endothelial cells, promoting monocyte engraftment [70]. Upon engraftment, monocytes acquired liver-associated transcription factors such as IDE3 and LXR-α, showing that the KC niche is composed of stellate cells, hepatocytes, and endothelial cells, which together imprint tissue-specific macrophage identity (Figure 1). Although both studies employed artificial systems for KC compartment depletion, it is plausible that these observations are similar to those observed in erythromyeloid progenitor (EMP)-to-KC differentiation during development.

### 4.2. Macrophages and Endothelial Cells in Liver Disease

Despite the tight regulation of different liver cell compartments, these aforementioned interactions may be disrupted and affected upon tissue damage and inflammation. Using partial hepatectomy in mice, it was shown that monocytes are recruited and infiltrate the liver during mass regeneration. These monocytes were also located adjacent to sprouting points [71], crucial for angiogenesis in endothelial cells [72]. Further, this interaction occurs primarily through local secretion of Wnt5a by monocytes. Moreover, following partial hepatectomy, CD11b KO mice showed reduced angiogenesis, liver mass regeneration, and survival [71]. Of note, the role of KCs in this process is only relevant at later time points after hepatectomy, suggesting that this interplay occurs between monocyte-derived macrophages and endothelial cells. In patients with inflammatory liver disease, sinusoidal endothelial cells (LSECs) were shown to recruit leukocytes via VAP1, an adhesion protein, found to be upregulated in the liver and serum of these patients [73]. LSECs are one of the first targets of injury during the inflammatory phase [74]. Albeit LSECs are known to release ROS during inflammation, they act as antioxidants upon LPS activation to counteract the increased amounts of ROS produced by KCs [75]. In general, activation of endothelial cells during liver injury mainly promotes leukocyte adhesion to the liver endothelium [74] (Figure 1). An in vitro study reported monocytes’ ability to regulate endothelial function by secreting angiopoietin-1 (Ang-1), a growth factor that induces Tie2 phosphorylation in endothelial cells, promoting survival particularly under low nutrient and cytokine stress conditions [76]. In a different in vitro study, it was demonstrated that human monocytes bind directly to injured endothelial cells, inducing their proliferation and migration. Here, the authors found that cells enter S-phase 12–16 h after the addition of primary monocytes. This mechanism occurs first through a direct crosstalk between monocytes and endothelial cells, followed by HGF-Met signaling [77].

On another hand, endothelial cells can also modulate monocytes/macrophages. Schubert et al. observed in vitro that in presence of macrophage colony-stimulating factor (M-CSF), endothelial cells were able to stimulate the proliferation of monocytes after 6 days in culture [78].

One of the hallmarks of liver disease is the initiation of an inflammatory process characterized by recruitment of monocyte-derived macrophages to the liver. In NASH, LSEC activation induces their acquisition of an inflammatory profile, translating into KC activation and thus propagation of the inflammatory milieu [79]. Progression from simple steatosis to NASH has been shown to compromise LSEC function, leading to KC and hepatic stellate cell (HSC) activation [52] (Figure 1). Moreover, it was recently reported that NASH diet in mice induced significant changes in KC enhancers and gene expression, therefore causing partial loss of this compartment [80]. These changes were accompanied by induction of *Trem2* and *Cd9* expression. KC loss was compensated by infiltration of monocyte-derived macrophages that exhibited convergent epigenomes, transcriptomes, and functions to KCs [80]. Intriguingly, these NASH-induced changes in KCs promoted a scar-associated macrophage phenotype. In agreement with literature findings emphasizing the importance of KC and LSEC communication under homeostatic conditions [69,70], the authors found that KCs were spatially closer to DLL4+ CD138+ LSECs as compared to recruited macrophages. This supports a role for LSECs as promoting niche specialization and, in the setting of liver disease such as NASH, determining the molecular phenotypes of KCs derived from recruited macrophages (Figure 1). Another study emphasizes the interaction between LSECs and recruited myeloid cells in a mouse model of obesity with hepatic inflammation and glucose intolerance, showing that blockade of VLA-4 on LSECs reduced leukocyte recruitment and ameliorated hepatic inflammation [57].

Previous studies have suggested crosstalks between macrophages and endothelial cells undergoing EndMT [81,82], including an experimental model of atherosclerosis that revealed that macrophages partially induce this transition [81]. In addition, the EndMT process may modulate macrophage differentiation through secretion of soluble factors, as EndMT-conditioned media significantly decreased macrophage proliferation.

### 4.3. Trem-2 as a Likely Mediator of Macrophage/Monocyte-Endothelial Cell Interactions

Triggering receptor expressed on myeloid cells-2 (Trem-2) is an immune receptor initially described as being expressed in myeloid cells such as monocytes, neutrophils, dendritic cells, macrophages, microglia, and osteoclasts as well as on megakaryocytes and platelets [83,84,85]. Trem-2 acts as a dampener of the inflammatory response, contributing, among other actions, to control the magnitude of macrophage activation [86]. Functional activation driven by Trem-2 ligation has been shown to involve different pathways, culminating in the control of various cellular processes including cell maturation, phagocytosis, cell proliferation and survival, and regulation of pro-inflammatory mediators [87]. In the past few years, it has been proposed that Trem-2 is also expressed in some cell types from the non-myeloid lineage, namely, adipocytes [88] and hepatic stellate cells in the liver [89].

Trem-2 ligands leading to macrophage activation in situ have not been identified but different studies reported the binding to phospholipids such as phosphatidylserine [90,91] and a range of acid and zwitterionic lipids [92], which may accumulate upon cell death. In addition, Trem-2 has been shown to bind a wide array of anionic molecules such as bacterial products, DNA, and lipoproteins [87]. Interestingly, Trem-2 ligands may be present under physiological or pathophysiological conditions, showing that Trem-2 activation and regulation is a complex process and which depends on the environmental context. Moreover, some reports suggest that activation of Trem-2 induces Trem-2 ligands production, therefore sustaining Trem-2-dependent activation [93,94].

As such, the importance of Trem-2 in the context of liver disease has gained great attention. Perugorria et al. demonstrated that mice lacking Trem-2 exhibited increased liver damage and inflammation both in acute (APAP-induced) and chronic (CCl4-induced) liver damage mouse models [89]. Trem-2 was expressed in KCs and in hepatic stellate cells (HSC), indicating that its expression on non-parenchymal cells functions as a natural break on inflammation. Trem-2 expression was also increased in livers from cirrhotic patients [89]. Similarly, compared to non-tumoral tissue, Trem-2 expression was increased in tumors from both mice and patients with hepatocellular carcinoma (HCC) [95]. Furthermore, genetic ablation of Trem-2 augmented tumor development and exacerbated liver damage and inflammation. Specifically, Trem-2 was found to be primarily expressed in infiltrating macrophages [95] (Figure 2).

Single-cell RNA-sequencing analyses of non-parenchymal cells from healthy and NASH murine livers displayed distinct macrophage and endothelial cell population clusters. Endothelial cells from NASH livers revealed increased expression of genes involved in lipid metabolism, antigen presentation, and chemokine release, while genes involved in vascular development and homeostasis showed decreased expression. Moreover, endothelial receptors and angiocrine factors were downregulated during diet-induced NASH. In the same setting, unique macrophage clusters were identified, and of great interest was the identification of two KC populations with low and high levels of Trem2 mRNA expression. Significantly, 93% of Trem2hi KCs were derived from NASH livers, supporting their unique high association with steatohepatitis and hence named “NASH-associated macrophages” (NAMs) [96]. NAMs also expressed CD9 and Gpnmb. Gpnmb mRNA expression has been previously associated with liver injury and fibrosis [97].

Strikingly, NAMs were identified in NASH patients, where hepatic Trem2 expression was strongly associated with the severity of steatosis, inflammation, hepatocyte ballooning, liver fibrosis, and NAFLD activity score [96] (Figure 2).

Likewise, a scar-associated macrophage population expressing Trem-2 and CD9 was identified in cirrhotic livers [98]. These macrophages were differentiated from circulating monocytes and showed pro-fibrogenic properties. Surprisingly, in the same study, a particular endothelial cell population characterized by the expression of atypical chemokine receptor 1 (ACKR1) and plasmalemma vesicle-associated protein (PLVAP) was also identified. These expanded during fibrosis and enhanced leukocyte transmigration. These cells were topographically associated with the fibrotic niche. Various paracrine and autocrine interactions were detected between their ligands and endothelial cell cognate receptors.

In line with Perugorria et al. [89], we have recently shown that in acute and chronic mouse models of liver disease, Trem-2 is particularly upregulated in a specific macrophage population, named “transition macrophages” [99]. Transcriptomic analyses showed that absence of Trem-2 impacts the switching from recruited to transition macrophages, with the latter showing muted transcriptional response to oxidative stress and an inability to shut down the inflammatory profile. Moreover, Trem-2 expression allowed an adequate replenishment of the KCs compartment. Interestingly, an endothelial liver damage endothelial cell (LDECs) population showing a distinct transcriptional profile was identified. The accumulation of LDECs significantly correlated to the degree of liver damage and to impaired replenishment of KC compartment [99] (Figure 2). These results fit the notion that a crosstalk between macrophages and endothelial cells is part of the liver regeneration process after liver injury.

Taken together, these studies highlight the relevance of cell-to-cell interaction, namely, macrophages and endothelial cells during liver disease. Trem-2 appears as important mediator of these interactions possibly by fine tuning inflammatory responses and simultaneously controlling the extent of endothelial cell activation.

## 5. Conclusions and Future Perspectives

There is a growing interest in delineating liver cell crosstalk-specific mechanisms. Remarkably, alterations in cell-to-cell communication are observed throughout the spectrum of hepatic disease, from simple steatosis to cirrhosis. Characterizing these cell-to-cell interactions will (1) allow a deeper understanding of liver biology, by highlighting the tightly regulated processes responsible for tissue integrity maintenance and (2) pin point key pathways that may be altered or affected during liver disease, potentially constituting promising targets for pharmacological interventions and adjuvant therapies.

Here, we reviewed the current understanding of macrophage/monocyte and endothelial cell communication in the liver. It is now most evident that, rather than fixing on a single-cell population, targeting these interactions among populations may offer a more effective control and treatment of pathogenesis.

Of particular interest, we highlighted the role of Trem-2 in mediating these macrophage–endothelial cell interactions. Given recent findings that Trem-2 macrophage expression is instrumental in regulating inflammation and in determining severity of hepatic damage, agonist/antagonist applications of the receptor may provide a path towards enhanced hepatic regeneration via LSEC activation control. As Trem-2 expression impacts mostly on recruited macrophages, targeting the protein might prove clinically significant in the prominent inflammatory state of advanced disease. For these reasons, the molecular players and cellular processes involved in this crosstalk mediated by Trem-2 warrant further investigations.

In summary, evaluating interlinks between macrophage and endothelial cells will expand our knowledge of hepatic pathogenesis and guide novel systemic therapeutic strategies for controlling pathogenesis and promoting tissue repair and disease regression.

## Figures and Tables

**Figure 1 jcm-10-01248-f001:**
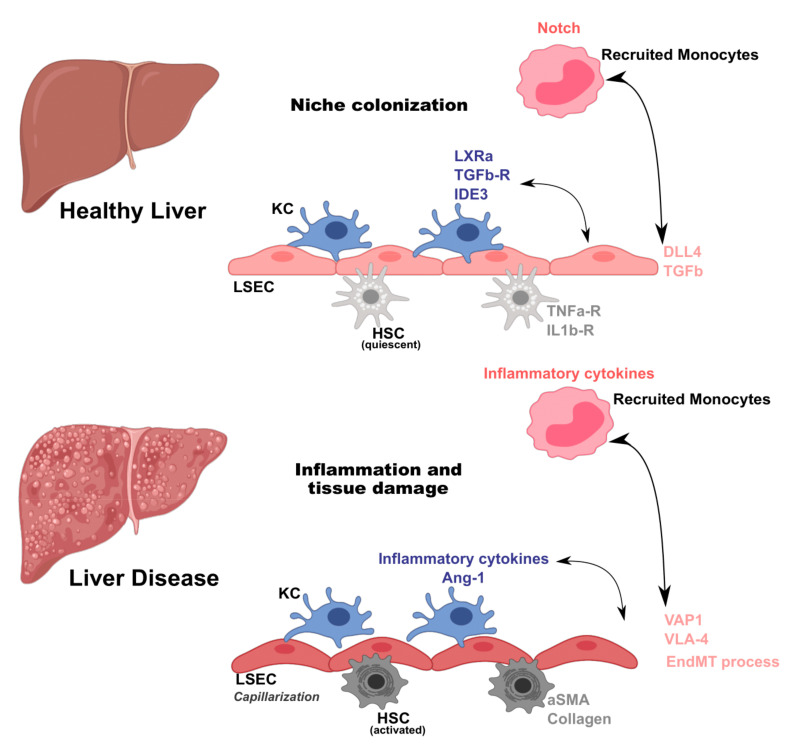
Macrophage and endothelial cell crosstalk in homeostasis and liver disease.
Liver cell interactions occur by tightly regulated mechanisms. In homeostatic conditions, perturbations in the Kupffer cell (KC) niche induce recruitment of monocytes. Once in the liver, monocytes receive signals from liver sinusoidal endothelial cells (LSEC) such as delta like canonical notch ligand 4 (DLL4) and transforming growth factor beta (TGFb), and from hepatic stellate cells (HSC) through TNFa-R and interleukin receptor 1 receptor (IL1b-R), that result in acquisition of KC-specific transcriptional program (expression of liver X receptor alpha (LXRa), transforming growth factor beta (TGFb-R), and inhibitor of DNA binding 3 (IDE3)). During liver disease, macrophages and endothelial cells are activated and their interactions amplify inflammation and tissue damage. Macrophages secrete inflammatory cytokines and chemokines and growth factors (Ang-1), while endothelial cells increase expression of adhesion molecules (vascular adhesion protein 1-VAP1 and integrin alpha-4 subunit-VLA-4). Moreover, infiltrating macrophages induce endothelial to mesenchymal transition (EndMT) on endothelial cells, a key process for fibrosis induction and establishment in liver pathology, and hepatic stellate cell (HSC) activation that secrete α-smooth muscle actin (α-SMA) and collagen.

**Figure 2 jcm-10-01248-f002:**
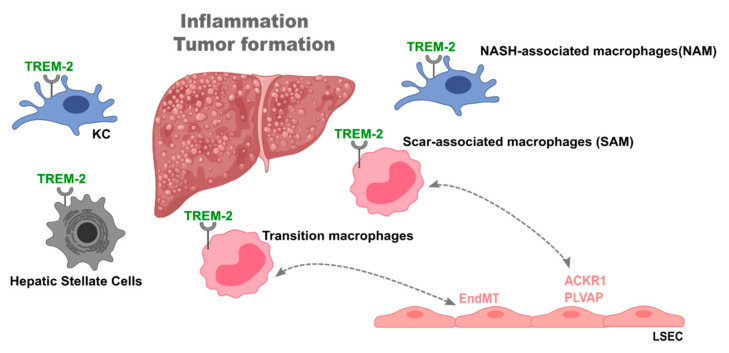
The role of triggering receptor expressed on myeloid cells-2 (Trem-2) in liver disease. Trem-2 plays a fundamental role during liver disease, namely, in non-alcoholic steatohepatitis (NASH), fibrosis, and cirrhosis. Liver Trem-2 expression was shown to control tumor formation and inflammation and being increased in livers of cirrhotic patients. Trem-2 is expressed in different cell types such as Kupffer cells, recruited macrophages, and hepatic stellate cells. In particular, different macrophage populations, i.e., NASH-associated macrophages (NAM), scar-associated macrophages (SAM), and transition macrophages, are characterized by Trem-2 expression and found in specific settings of liver disease. Further, SAM and transition macrophages interact with a particular endothelial cell population expressing atypical chemokine receptor 1 (ACKR1) and plasmalemma vesicle-associated protein (PLVAP) and induce an endothelial to mesenchymal transition, respectively, in the disease setting. This places Trem-2 as plausible mediator of liver macrophage-endothelial cell interactions.

## Data Availability

Not applicable.

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
