# Peer review of "Insights into Macrophage/Monocyte-Endothelial Cell Crosstalk in the Liver: A Role for Trem-2"

_jcm, 2021, doi:10.3390/jcm10061248_

Round 1

Reviewer 1 Report

Authors have nicely described that macrophages and endothelial cells are pivotal players in promoting and sustaining disease progression. However, the provided Cartoon should be optimized namely the appearance of fenestrated Endothelial cells and the activated hepatic stellate cells in diseased state.

Author Response

Dear Reviewer 1,

Thank you for your comments and suggestions. I have revised the cartoon (figure 1) and integrated your suggestions.

Reviewer 2 Report

Title should refer to monocyte/macrophage-endothelial cell crosstalk - as many examples relate to monocyte endothelial cell interactions in Section 4.

Clearly Trem-2 expression defines distinct populations of monocyte/macrophages but there is no discussion of a potential role of Trem-2 in cell to cell interactions or via cognate recognition of a ligand.  I understand that the ligands for Trem-2 are not defined but this should be mentioned.

I have marked up the PDF highlighting in yellow minor typo's - but also the occasional phrase which is hard to understand and could be rewritten. I also did a suggested edit on the Figure 1 legend.

Author Response

Dear Review 2,

Thank you for your comments and suggestions.

I changed the title to macrophage/monocyte.

I acknowledge your suggestion and introduced a paragraph referring to Trem-2 and its ligands (please see line 348).

I am grateful for your suggestions and have rewritten some sentences that were highlighted throughought the text (please see “track changes”)
